# Decoding the Function of *FGFBP1* in Sheep Adipocyte Proliferation and Differentiation

**DOI:** 10.3390/ani15101456

**Published:** 2025-05-18

**Authors:** Liming Tian, Zhaohua He, Guan Wang, Shuhong Zhang, Tenggang Di, Menghan Chang, Wei Han, Jingyi Gao, Meng Li, Ziyi Wang, Huan Zhang, Shaobin Li, Guangli Yang

**Affiliations:** 1College of Biology and Food, Shangqiu Normal University, Shangqiu 476000, China; liming202506@163.com (L.T.); hezh1250148718@163.com (Z.H.);; 2Gansu Key Laboratory of Herbivorous Animal Biotechnology, Faculty of Animal Science and Technology, Gansu Agricultural University, Lanzhou 730070, China; 3College of Animal Science and Technology, Henan Agricultural University, Zhengzhou 450046, China

**Keywords:** adipocyte differentiation, adipocyte proliferation, *FGFBP1*, preadipocytes, sheep, tail fat deposition

## Abstract

Tail adipose deposition is a crucial economic trait in sheep with fat-tailed and fat-rumped characteristics. This study evaluated the regulatory function of the *FGFBP1* gene in the proliferation and differentiation of ovine tail preadipocytes based on gene overexpression and knockdown experiments. *FGFBP1* overexpression significantly inhibited cellular proliferation and promoted adipocyte differentiation, whereas gene silencing led to the opposite effects. These findings confirm *FGFBP1* as a key regulator of ovine tail adipogenesis. This study offers a novel understanding of the mechanisms of adipose tissue development and identifies a potential molecular target for optimizing tail adipose deposition in sheep.

## 1. Introduction

Sheep (*Ovis aries*), as one of the earliest domesticated species, have significantly contributed to human society through their provision of wool and meat resources [1,2]. Tail morphology, a critical breed characteristic influencing economic outcomes in fat-tailed and fat-rumped sheep husbandry, is classified into the following five categories based on the fat deposition patterns of sheep and their physical conformations: fat-rumped, short–fat-tailed, long–fat-tailed, short–thin-tailed, and long–thin-tailed [3]. Genetic studies substantiate that modern fat-tailed sheep lineages evolved from ancestral thin-tailed breeds via natural selection and artificial breeding [4,5]. These breeds are globally valued in livestock production for their robust environmental adaptability, disease resistance and high productivity, all of which can be further enhanced under optimal nutritional conditions [6,7]. Fat composition and metabolic regulation determine the flavor profiles and nutritional attributes of mutton products [8]. At the biological level, lipid metabolism networks not only maintain systemic homeostasis in sheep but also play critical roles in nutrient assimilation and physiological health [9,10,11]. However, excessive tail adipose deposition leads to a decrease in the market value by compromising meat quality and inducing reproductive dysfunction. From the perspective of animal production, this condition further reduces the feed conversion efficiency, adversely affecting economic returns [3,11]. Increasing consumer demand for healthier meat products has increased the market preference for low-fat mutton. Although tail docking transiently mitigates caudal fat accumulation, it promotes lipid deposition in subcutaneous and visceral compartments, ultimately degrading the meat quality and feed utilization efficiency [12,13]. This necessitates the identification of genetic regulators and molecular markers associated with tail adipogenesis in sheep.

Fibroblast growth factor-binding protein 1 (*FGFBP1*) is a secreted chaperone that mobilizes extracellular matrix-bound fibroblast growth factors (FGFs) and facilitates the activation of their receptors. Emerging evidence indicates that *FGFBP1* functions as a crucial amplifier of FGF signaling cascades [14], which are known to regulate adipocyte development via multiple pathways. While *FGFBP1* has been implicated in developmental processes such as angiogenesis and neural differentiation [14,15,16,17], its specific role in adipocyte biology is poorly understood. This knowledge gap motivated our investigation into how *FGFBP1* modulates ovine adipose proliferation and differentiation.

Transcriptomic data analysis of tail adipose tissues from Large-tailed Han sheep and Small-tailed Han sheep suggests that *FGFBP1* may play a critical role in tail adipose deposition (unpublished). To elucidate its regulatory mechanisms in ovine tail adipogenesis, we employed Real-Time Quantitative PCR (RT-qPCR), Cell-Counting Kit-8 (CCK-8) cell proliferation assays, 5-ethynyl-2′-deoxyuridine (EdU) staining, and Oil Red O lipid staining to systematically evaluate the effects of *FGFBP1* on the proliferation and differentiation of tail-derived adipocytes.

## 2. Materials and Methods

### 2.1. Cell Culture

Cryopreserved preadipocytes were thawed at 37 °C in a water bath, mixed with complete medium (Dulbecco’s Modified Eagle Medium/F12, 10% fetal bovine serum, 1% penicillin–streptomycin), centrifuged at 2500 rpm for 3 min, resuspended, and cultured at 37 °C in an environment of 5% CO_2_.

### 2.2. Cell Differentiation

Preadipocytes were cultured in 6-well plates until 80% confluence, then induced with differentiation medium A (1.72 μmol/L insulin, 5.10 μmol/L dexamethasone, 0.122 mmol/L IBMX) for 48 h. Cells were subsequently subjected to RT-qPCR analysis. The medium was subsequently replaced with differentiation medium B (1.72 μmol/L insulin) every 48 h for a 12-day adipogenic induction period.

### 2.3. Cell Transfection

The old medium was replaced with fresh complete medium to ensure optimal nutrient supply. Cell transfection was performed using the Lipo8000™ transfection kit (Beyotime Biotechnology, Shanghai, China) following the manufacturer’s instructions. After transfection, cells were maintained under standard culture conditions (37 °C, 5% CO_2_) for 24–48 h before harvesting for subsequent experimental analyses. The transfection plasmids, including pcDNA3.1, pcDNA3.1-*FGFBP1*, si-NC, and si-*FGFBP1*, were obtained from GenePharm (Shanghai, China).

### 2.4. Real-Time Quantitative PCR

Total RNA was isolated from cultured cells, and complementary DNA (cDNA) was synthesized using the PrimeScript™ RT Kit (TaKaRa, Dalian, China) in accordance with the manufacturer’s protocol. Gene-specific primers for qPCR were designed with DNAMAN 6.0 software (Table 1). RT-qPCR amplification was conducted using SYBR Green PCR Master Mix (TaKaRa, Dalian, China) on a QuantStudio™ 5 Real-Time PCR system. The relative gene expression levels were quantified using the 2^−ΔΔCt^ analytical method, with *β-actin* serving as an endogenous control for data normalization. Three technical replicates were performed for each group (*n* = 3).

### 2.5. Cell Proliferation Analysis

Cell viability was evaluated using the CCK-8 (Beyotime, Shanghai, China), and the cell proliferation quantity was determined by the EdU assay (Beyotime, Shanghai, China). Cells were seeded in 96-well plates with six independent replicates per group. At 12, 24, 36, and 48 h after culture, 10 μL of CCK-8 reagent was added to each well and incubated at 37 °C for 1 h. Absorbance was measured at 500 nm using a microplate reader (Perkin Elmer, Singapore). Cell proliferation was also assessed using the EdU assay using the BeyoClick™ EdU-594 Cell Proliferation kit (Beyotime, Shanghai, China) according to the manufacturer’s protocol. Three technical replicates were performed for each group (*n* = 3).

### 2.6. Cell Cycle Analysis

Overexpression and knockdown experiments were initiated when the cells reached 80% confluency. After 36–48 h of treatment, the cells were harvested by trypsinization for subsequent analysis. All experimental procedures were performed in strict accordance with the manufacturer’s protocols provided in the Cell Cycle and Apoptosis Detection kit (Beyotime, Shanghai, China).

### 2.7. Oil Red O Staining

After fixing with 4% paraformaldehyde (PFA) for 20 min, Oil Red O staining was performed using a commercially obtained staining kit (Beyotime, Shanghai, China) to visualize lipid deposition. Quantitative analysis was conducted by measuring the absorbance at 500 nm using a spectrophotometer.

### 2.8. Western Blotting

Cellular proteins were extracted using a protein extraction kit (Beyotime, Shanghai, China). Protein concentrations were determined using a bicinchoninic acid assay kit (Beyotime, Shanghai, China). Protein samples were denatured by boiling in sodium dodecyl sulfate (SDS) loading buffer at 97 °C for 7 min and subsequently loaded onto 12% SDS-PAGE gels for electrophoretic separation. The separated proteins were electrophoretically transferred to polyvinylidene fluoride (PVDF) membranes using a wet transfer system. After the transfer, the membranes were blocked with 5% skim milk in TBST buffer for 1.5 h at room temperature. Next, the membranes were sequentially incubated overnight with the primary antibodies *FGFBP1* Rabbit pAb (Catalog #BS-1768R, Bioss Antibodies, Beijing, China) at 4 °C and horseradish peroxidase-conjugated secondary antibodies for 1 h at room temperature. *β-actin* was used as an internal control for normalization purposes. Protein bands were visualized using BeyoECL Star chemiluminescent detection reagent (Beyotime, Shanghai, China).

### 2.9. Statistical Analysis

The data are presented as the mean ± standard deviation. The SPSS 27 software package was used for statistical analysis. The experimental group served as independent variables, while the measured value was the dependent variable. The data were analyzed using an independent-samples *t* test. The levels of significance were as follows: ns (*p* > 0.05), * *p* < 0.05, ** *p* < 0.01, and *** *p* < 0.001. GraphPad Prism 9.5 was used for data visualization.

## 3. Results

### 3.1. Determination of FGFBP1 Overexpression and Knockdown Efficiency

Cells were transfected with either overexpression constructs (pcDNA3.1-*FGFBP1* and empty pcDNA3.1 vector) or siRNA oligonucleotides (specific si-*FGFBP1* and non-targeting si-NC) to determine the biological function of *FGFBP1* in preadipocytes. RT-qPCR demonstrated that the *FGFBP1* mRNA levels were significantly elevated in the overexpression group compared with those in the empty vector control group (*p* < 0.001) (Figure 1A). Conversely, *FGFBP1* knockdown resulted in a marked reduction in mRNA levels relative to the non-targeting control group (*p* < 0.05) (Figure 1B). Western blotting confirmed the dose-dependent upregulation of the FGFBP1 protein in the overexpression group and its corresponding downregulation in the knockdown group (Figure 1C,D). These findings validate the successful establishment of the *FGFBP1* overexpression and knockdown models, providing a foundation for subsequent functional studies.

### 3.2. Effects of FGFBP1 on the Proliferation of Sheep Preadipocytes

#### 3.2.1. Determination of Proliferation Marker Genes After *FGFBP1* Overexpression and Knockdown

The expression patterns of cell proliferation marker genes (*CCNB1*, *Cyclin B3*, *Cyclin D*, and *PCNA*) in response to *FGFBP1* modulation were systematically analyzed. *FGFBP1* overexpression significantly downregulated *PCNA* mRNA expression compared with that in the control group, whereas *CCNB1* and *Cyclin D* exhibited nonsignificant downward trends (Figure 2A). Conversely, *FGFBP1* knockdown induced a significant upregulation of *CCNB1*, *Cyclin D*, and *PCNA* expression, with *Cyclin B3* levels remaining unaltered (Figure 2B). Notably, changes in *PCNA* expression, a core component of the DNA replication complex, showed a significant association with alterations in the cell proliferation status.

#### 3.2.2. CCK-8 Assay Results After *FGFBP1* Overexpression and Knockdown

The viability of preadipocytes was quantified using the CCK-8 assay. After *FGFBP1* overexpression, a statistically significant reduction in cell viability was noted at both the 12 and 48 h time points relative to the corresponding controls (Figure 3A). Conversely, *FGFBP1* knockdown led to a progressive increase in cell viability under similar experimental conditions (Figure 3B).

#### 3.2.3. EdU Assay Results After *FGFBP1* Overexpression and Knockdown

The EdU assay was used to further evaluate the effects of *FGFBP1* on cell proliferation. The fluorescence intensity was determined using a microplate reader approximately 24 h after transfection. A significant increase in the number of proliferating cells was noted in si-*FGFBP1*-transfected cells (Figure 4B), whereas no statistically significant changes in proliferation were observed in cells transfected with pcDNA3.1-*FGFBP1* (Figure 4A).

### 3.3. Effects of FGFBP1 Overexpression and Knockdown on the Cell Cycle

Flow cytometry was used to determine the effects of *FGFBP1* on cell cycle progression. *FGFBP1* overexpression did not induce significant changes in the G1 phase population, but resulted in a decreased proportion of cells in the S phase and an accumulation of cells in the G2 phase compared to that in the control group (Figure 5A–C). These findings suggest that *FGFBP1* overexpression promoted cell cycle arrest in the G2/M phase, suppressing cellular proliferation. Conversely, *FGFBP1* knockdown led to no notable changes in the G1 phase population, but a significant reduction in the G2 phase population along with an increase in the S phase fraction population were noted (Figure 5D–F). This finding indicates accelerated DNA replication and premature entry into mitosis, collectively demonstrating an enhanced proliferative capacity. Thus, *FGFBP1* overexpression inhibited cellular proliferation, whereas *FGFBP1* knockdown alleviated this inhibitory effect.

### 3.4. Effects of FGFBP1 on the Differentiation of Sheep Preadipocytes

#### 3.4.1. Effects of *FGFBP1* Overexpression and Knockdown on Cell Differentiation

Gain- and loss-of-function experiments were conducted using *FGFBP1* overexpression and knockdown to elucidate the regulatory role of *FGFBP1* in ovine preadipocyte differentiation. The expression dynamics of key adipogenic markers, including *PPARγ*, *Adiponectin*, *C/EBPα*, and *FABP4*, were systematically monitored at 48 h. *FGFBP1* overexpression significantly upregulated the *PPARγ* and *FABP4* transcripts (Figure 6A), whereas *FGFBP1* knockdown substantially suppressed the expression of *PPARγ*, *Adiponectin*, *C/EBPα*, and *FABP4* (Figure 6B).

#### 3.4.2. Oil Red O Staining Analysis After *FGFBP1* Overexpression and Knockdown

Oil Red O was used to stain lipid droplets and assess the adipogenic differentiation of preadipocytes. *FGFBP1* overexpression significantly promoted lipid droplet accumulation in preadipocytes (Figure 7A–C). Conversely, *FGFBP1* knockdown markedly reduced the lipid droplet content (Figure 7D–F). These findings indicated the critical regulatory role of *FGFBP1* in preadipocyte differentiation. Changes in its expression levels directly modulated the extent of lipid droplet accumulation, thereby influencing adipocyte differentiation.

## 4. Discussion

This study is the first of its kind to present evidence of the dual regulatory function of *FGFBP1* in modulating the proliferation and differentiation processes in ovine tail adipocytes. *FGFBP1* knockdown significantly upregulated the expression of the proliferation-associated genes (*CCNB1*, *Cyclin D*, and *PCNA*) in preadipocytes, whereas its overexpression suppressed the levels of these proliferative markers. These findings were in accordance with the S-phase arrest determined using flow cytometry and were validated using complementary CCK-8 and EdU assays. These assays collectively confirmed the inhibitory role of *FGFBP1* in preadipocyte proliferation. Notably, while genetic manipulation elicited significant transcriptional effects, the corresponding protein-level alterations were undetected in this study. Therefore, subsequent studies should use western blotting to elucidate the potential post-transcriptional regulatory mechanisms.

As a member of the *FGFBP* family [17,18], *FGFBP1* functions as a critical modulator of FGF signaling [19,20,21]. Its capacity to bind multiple FGF ligands (including FGF1, FGF2, FGF7, FGF10, and FGF22) enhances FGF receptor interaction and subsequent signal transduction [22]. Several studies have established *FGFBP1* as an oncogenic factor within the tumor microenvironment by virtue of its activation of proliferative FGFR signaling pathways [23,24,25,26], a mechanism consistent with the well-documented pro-proliferative effects observed in various cancer models [27,28]. In contrast, our finding uncovered a distinct biological role of *FGFBP1* in adipocytes, demonstrating its potential to regulate adipocyte differentiation through the preferential activation of FGFR subtypes associated with differentiation control. This tissue-dependent functional dichotomy may arise from the selective engagement of microenvironment-specific FGFR isoforms by *FGFBP1*, as supported by previous findings [22,29,30]. Furthermore, our findings revealed that *FGFBP1* overexpression significantly upregulated *PPARγ* expression, which was contradictory to the established adipogenesis inhibition mediated by Twist1 [31]. We hypothesized that *FGFBP1* might circumvent the inhibitory influence of Twist1 through activation of the PI3K/AKT-mTOR signaling cascade, thereby directly enhancing the transcriptional activation of adipocyte-specific genes.

In contrast to its antiproliferative activity, *FGFBP1* exhibits potent pro-differentiation effects. Knockdown attenuated key adipogenic markers (*PPARγ*, *Adiponectin*, *C/EBPα*, *FABP4*) and lipid droplet formation, whereas overexpression amplified differentiation markers and lipid accumulation. Temporal discrepancies between the gene expression patterns and Oil Red O staining outcomes suggested that lipid droplet maturation required the accumulation of differentiation regulators at a certain threshold. From the perspective of meat science, balanced adipocyte differentiation facilitates optimal intra-muscular fat distribution, whereas excessive proliferation may induce connective tissue hyperplasia [32,33,34,35]. The dual regulatory capacity of *FGFBP1*—simultaneously curbing proliferation while enhancing differentiation—could critically influence the meat quality of sheep by modulating the intramuscular versus tail fat ratio, thereby affecting tenderness and the deposition of flavor compounds. Despite these mechanistic insights, several paradoxes warrant resolution. First, the concurrent enhancement of proliferation and the impairment of differentiation after knockdown implies potential stage-specific regulatory functions that are reminiscent of developmental gene expression patterns [36]. Second, the pro-adipogenic activity of *FGFBP1* contradicts the antiadipogenic effects of classical FGF signaling [37,38], possibly reflecting the selective mobilization of specific FGF isoforms rather than the activation of canonical mitogenic pathways.

From the viewpoint of livestock breeding applications, the current study presents a groundbreaking perspective regarding the development of low-adiposity sheep varieties. The targeted modulation of *FGFBP1* expression can fine-tune caudal adipose deposition, potentially addressing the challenges of ectopic lipid accumulation associated with conventional tail docking practices [39,40], demonstrating substantial practical applications. However, critical consideration must be given to the pleiotropic functions of the gene in regulating angiogenesis and systemic metabolism [15,41,42], which may exert collateral effects on multiple physiological processes. Consequently, further studies should prioritize the development of transgenic animal models to comprehensively assess the longitudinal impacts on myofiber development and meat characteristics for the thorough evaluation of its potential use in breeding applications while maintaining genomic safety.

## 5. Conclusions

This study elucidates the dual regulatory function of *FGFBP1* in ovine caudal adipose tissue dynamics, demonstrating inhibitory effects on preadipocyte proliferation and promoting adipocyte differentiation. From an agricultural standpoint, the strategic modulation of *FGFBP1* expression may enable the precise regulation of ovine adipose distribution by selectively reducing caudal fat deposition while maintaining intramuscular adipogenesis, a critical determinant of meat palatability and market value. This molecular approach presents a potential alternative to traditional tail-docking practices, thereby circumventing the adverse effects associated with conventional physical interventions on lipid homeostasis. However, the pleiotropic nature of *FGFBP1* signaling necessitates the comprehensive evaluation of potential off-target effects before practical implementation in a real-world setting.

## Figures and Tables

**Figure 1 animals-15-01456-f001:**
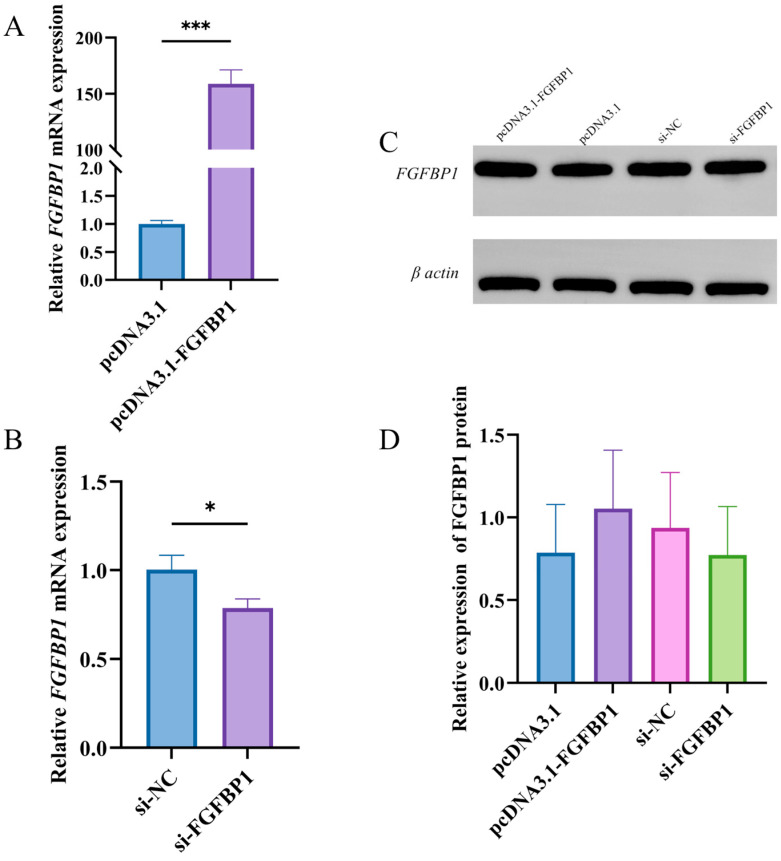
Validation of *FGFBP1* overexpression and knockdown efficiency in cultured cells. (**A**) Quantitative analysis of *FGFBP1* mRNA levels following genetic overexpression. (**B**) Assessment of *FGFBP1* mRNA reduction efficiency after targeted knockdown. (**C**,**D**) Protein validation after overexpression and knockdown. Statistical significance is denoted as follows: * *p* < 0.05, and *** *p* < 0.001.

**Figure 2 animals-15-01456-f002:**
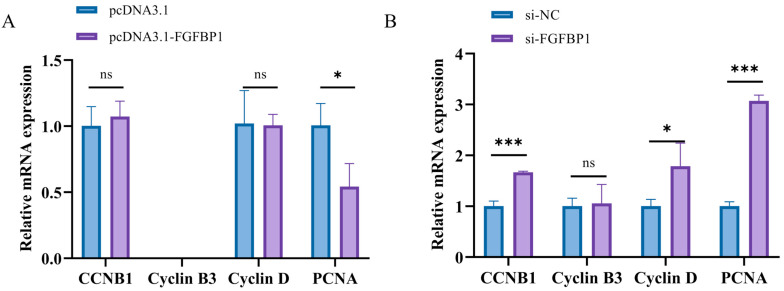
Proliferation of marker genes. (**A**) Effect of *FGFBP1* overexpression on proliferation marker genes. (**B**) Effect of *FGFBP1* knockdown on proliferation marker genes. Statistical significance is denoted as follows: ns (*p* > 0.05), * *p* < 0.05, and *** *p* < 0.001.

**Figure 3 animals-15-01456-f003:**
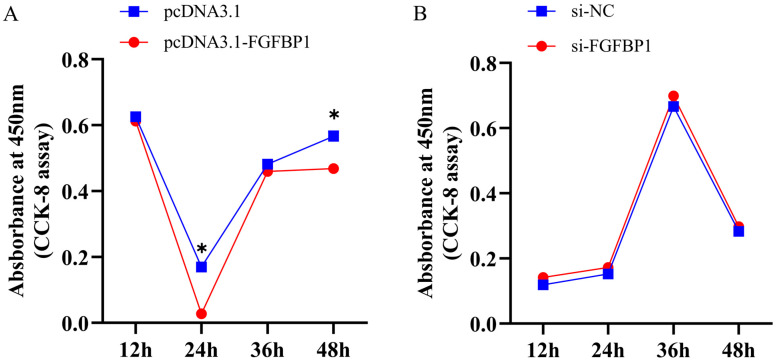
CCK-8 assay results following *FGFBP1* overexpression and knockdown. (**A**) Effect of *FGFBP1* overexpression on pre-adipocyte viability. (**B**) Effect of *FGFBP1* knockdown on pre-adipocyte viability. Statistical significance is denoted as follows: * *p* < 0.05.

**Figure 4 animals-15-01456-f004:**
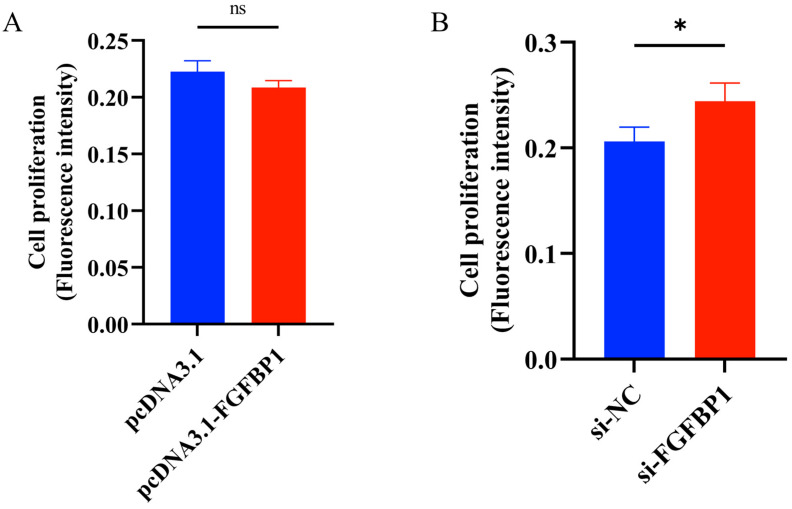
Changes in cell number following *FGFBP1* overexpression and knockdown as detected by EdU assay. (**A**) Cell proliferation following *FGFBP1* overexpression. (**B**) Cell proliferation following *FGFBP1* knockdown. Statistical significance is denoted as follows: ns (*p* > 0.05), * *p* < 0.05.

**Figure 5 animals-15-01456-f005:**
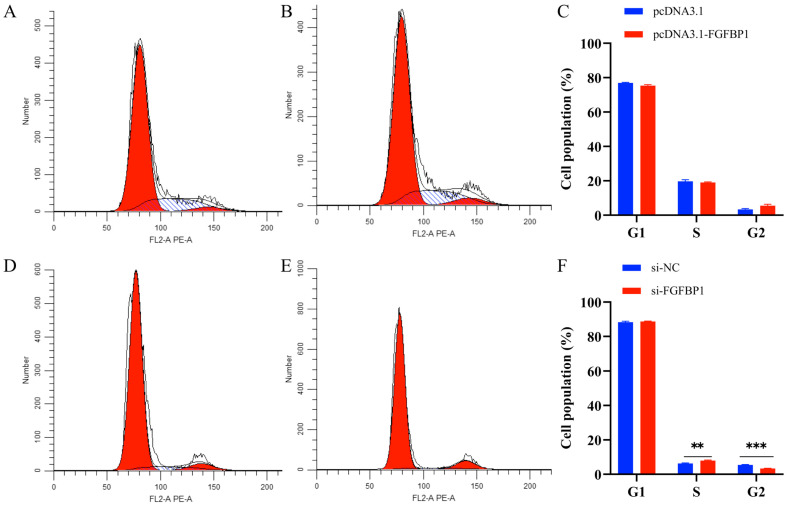
Flow cytometry analysis of cell cycle changes following *FGFBP1* overexpression and knockdown. (**A**) Cell cycle profile of pcDNA3.1-transfected cells. (**B**) Cell cycle profile of pcDNA3.1-*FGFBP1*-transfected cells. (**C**) Quantitative analysis of the cell cycle after *FGFBP1* overexpression. (**D**) Cell cycle profile of si-NC-transfected cells. (**E**) Cell cycle profile of si-*FGFBP1*-transfected cells. (**F**) Quantitative analysis of the cell cycle after *FGFBP1* knockdown. Statistical significance is denoted as follows: ** *p* < 0.01 and *** *p* < 0.001.

**Figure 6 animals-15-01456-f006:**
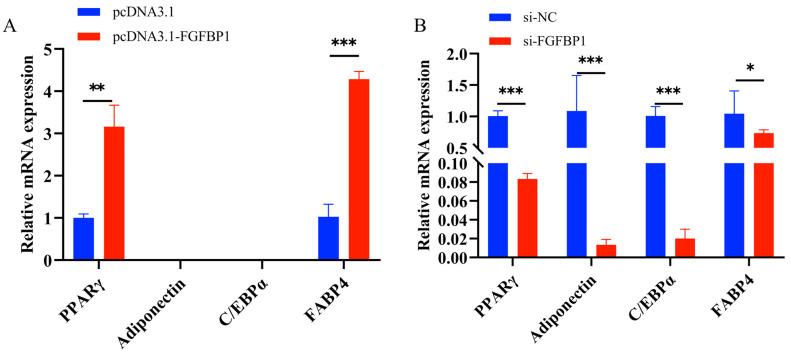
Changes in differentiation marker genes following *FGFBP1* overexpression and knockdown. (**A**) Effect of *FGFBP1* overexpression on differentiation marker genes. (**B**) Effect of *FGFBP1* knockdown on differentiation marker genes. Statistical significance is denoted as follows: * *p* < 0.05, ** *p* < 0.01 and *** *p* < 0.001.

**Figure 7 animals-15-01456-f007:**
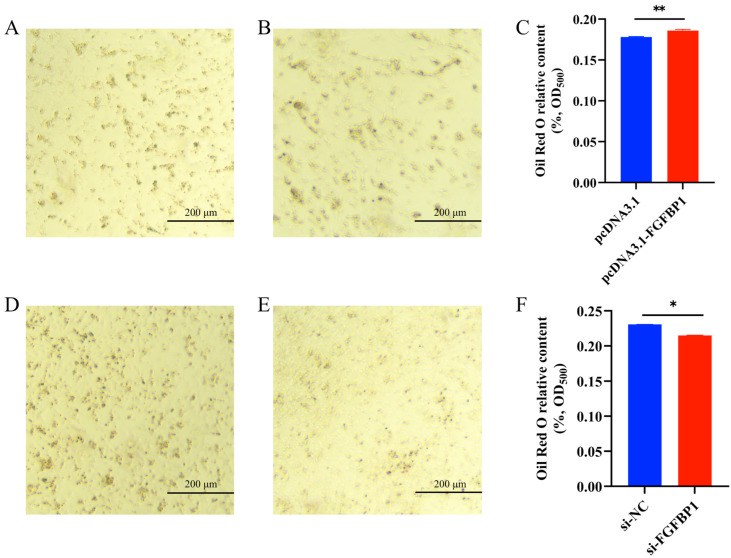
Results of Oil Red O staining after *FGFBP1* overexpression and knockdown. (**A**) Image of pcDNA3.1 transfection. (**B**) Image of pcDNA3.1-*FGFBP1* transfection. (**C**) Quantitative analysis of adipogenesis via spectrophotometric measurement of Oil Red O content at 500 nm wavelength. (**D**) Image of si-NC transfection. (**E**) Image of si-*FGFBP1* transfection. (**F**) Quantitative analysis of adipogenesis via spectrophotometric measurement of Oil Red O content at 500 nm wavelength. Statistical significance is denoted as follows: * *p* < 0.05 and ** *p* < 0.01.

**Table 1 animals-15-01456-t001:** RT-qPCR primers.

Gene	Gene ID	Primer Sequence (5′–3′)
*FGFBP1*	XM_004010004.5	F: CCTCCTCCTTCTGGCTGTTCTG R: TGCTTGGTTGGCTGGCTCCT
*CCNB1*	XM_027980034.1	F: CCCTCCAGAAATCGGTGACT R: AGCTCAACATCAACCTCTCCA
*Cyclin B3*	XM_012106700.4	F: GCTTGTCCAACACCGTCACCAT R: ACCTCCACCAACCAGTCCACAA
*Cyclin D*	NM_001127289.1	F: GGGATTGGGAGGTGCTGGTCTT R: AGGTCTGGGCGTGCTTCTTGA
*PCNA*	XM_004014340.5	F: GCTCAAGTGGCGTGAACCTACA R: TACGGTCGCAGCGGTAAGTGT
*PPARγ*	NM_001100921.1	F: TGCCGATTCCAGAAGTGCCTTG R: TCGCCCTCGCCTTTGCTTTG
*Adiponectin*	NM_001308565.1	F: AACCACTATGACGGCACCACTG R: ATAGAGGAGCACGGAGCCAGAG
*C/EBPα*	NM_001308574.1	F: ATGAGCAGCCACCTCCAGAG R: GCCAGGAACTCGTCGTTGAAG
*FABP4*	NM_001114667.1	F: AGGAAAGTGGCTGGCATGGC R: CTGGTAGCAGTGACACCGTTCA
*β-actin*	NM_001009784.3	F: GCAGGTCATCACCATCGGCAAT R: CGTGTTGGCGTAGAGGTCCTTG

## Data Availability

The datasets produced or examined in the course of this research are fully contained within the manuscript. The data supporting the conclusions of this study can be obtained from the corresponding author upon request.

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
