# Peer review of "Decoding the Function of *FGFBP1* in Sheep Adipocyte Proliferation and Differentiation"

_animals, 2025, doi:10.3390/ani15101456_

Round 1

Reviewer 1 Report

Comments and Suggestions for Authors

Dear Author,

Reviewer #MAJOR COMMENTS

The manuscript "Decoding the Function of FGFBP1 in Sheep Adipocyte Proliferation and Differentiation" is an original and interesting work. It is also important in its field of study. The methods presented are reliable, and the results are also well explained. However, the main issue with this MS is correlated with language, so the authors are strongly asked to be supported by a native speaker to enhance the language quality of this work. Try to make the paper more scientifically relevant by including content from more recent references. Keywords need to be alphabetically arranged.

Reviewer 2 Report

Comments and Suggestions for Authors

This study investigated the role of FGFBP1 in ovine tail fat development and identified its regulatory role in preadipocyte proliferation and differentiation, suggesting its potential as a molecular target for tail fat deposition in sheep. However, the interpretation of the results is somewhat limited, and the use of a small number of technical, biological, and experimental replicates weakens the strength and generalizability of the conclusions.

L60: Please remove the word “ectopic”

L64-77: As the manuscript is not intended to be a review, this section may be somewhat out of scope. You might consider removing or condensing it to improve clarity and relevance.

L80: Please include appropriate references to support

L86: It would be better to use the title “cell culture”

L87-97: The section is overly detailed. Please consider condensing it to avoid unnecessary, protocol-like descriptions. Also, please provide the composition of the complete culture medium. Please indicate how long adipogenic induction was performed.

L121: Does 'three biological replicates' mean using preadipocytes from three different sheeps, or does it refer to technical replicates. Please revise inequality symbols.

L134: Please clarify “designated treatment”

Figure3: Please remove the bar under the asterisk.

L210-L213: Please move to the discussion section.

Figure: Please remove “Note:”, provide more detailed explanation in figure legend, and indicate the name of each image.

Figure 5: Scale bars are required for clarity. The ORO-stained images are difficult to interpret. Please provide higher-quality images. Additionally, it is necessary to normalize lipid content by the number of cells.

Figure 7: Please use (A) instaed of Panel A.

Discussion : The main focus of this paper is the proliferation and differentiation of adipocytes which impact meat quality. It is not an study aimed at examining the relevance to human adipobiology. Please avoid the use of inappropriate abbreviations. The discussion section is overly unclear and incldues points that are not directly related to the results. It lacks a discussion of inconsistencies in the results when using different methods. Additionally, there is no analysis regarding the lack of differences at the protein level despite knockdown and overexpression at the gene level. The discussion section needs to be substantially rewritten

L300: Please include appropriate references to support

Reviewer 3 Report

Comments and Suggestions for Authors
  1. Where is Table 1?
  2. Figure 7, Where's the scale?
  3. Note the uniformity of gene italics throughout the text
  4. Figure 5 D,E,F .
  5. Please provide specific values and P-values of each group for significance analysis. (I doubt the authenticity of the existing significance differences, especially the existence of very significant labels, when the cell group changes by less than 5% in each period.)
  6. All of the figures lack an explanation for the significance mark in their notes.
  7. Please explain the source and species of antibodies to FGFBP1.
  8. Please provide identification photos (Marker stained) of the isolated and identified precursor adipocytes.

Author Response

请参阅附件。

Round 2

Reviewer 2 Report

Comments and Suggestions for Authors

L71-73: This reference is not written in English, so please remove the sentence (Zhang H, Zhang C, Di T G, et al. Differential Expression Analysis of Long Non-coding RNA in Tail Fat of Fat-tailed Sheep and Thin-tailed Sheep[J]. Journal of Henan Agricultural University, 2023, 57(2): 298-306. DOI:10.16445/j.cnki.1000-2340.20221107.001.)

L80: Please clarify if the ovine adipocytes are primary cells or cell lines. If it is primary cells, provide the isolation protocols and how to identify specific cell types. If it is cell lines, provide the source of cells.

L87: Please indicate the concentration using the correct unit (e.g., molarity).

L98: Please avoid using abbreviations in the title and also check if they have been already defined earlier in the text. For clarity, avoid abbreviations that are not repeatedly used.

L110: Please clarify the use of CCK-8. Is it for to assess cell viability, metabolic activity, or proliferation?

L117: If primary cells were used, it would be better to obtain cells from at least three different animals. If a cell line was used, it seems necessary to perform three independent replicate experiments.

L170: CCNB refers to cyclin B2? Please clarify why qPCR was performed separately for CCNB1 and CCNB.

Figure: I don't understand why there are so many figures in this paper. It would be helpful to separate and organize the data within the figures for proliferation and differentiation. Additionally, the font size varies between figures and even within the same graph. The overall graphs should be revised by referring to other published journal articles.

Figure 2A: It is difficult to understand how CCNB1 is expressed while Cyclin B is not.

Figure 5ABDE: It is difficult for readers to grasp what this figure is intended to represent.

Figure 6: In Figure 7, there are dots that appear to represent lipids, but in Figure 6A, adiponectin and C/EBPα are not expressed. Why is this the case? Were different samples with different differentiation time points used? The results do not match.

Figure 7: I still don't think the images in Figure 7 show ORO staining. The adipocytes do not appear to be fully differentiated. Additionally, if the data is not normalized to cell number, it would be difficult to prove whether the differences are due to increased adipogenesis or differences in cell number. Furthermore, there are differences in the lipid content of each control.

Reviewer 3 Report

Comments and Suggestions for Authors

The author has responded to each of my suggested revisions and made appropriate corrections to the images and text that needed to be modified. I believe it can be accepted in its current form.
